exKidneyBERT: a language model for kidney transplant pathology reports and the crucial role of extended vocabularies

Yang Tiancheng t77yang@uwaterloo.ca 1
Sucholutsky Ilia 1
Jen Kuang-Yu 2
Schonlau Matthias 1
1 Department of Statistics and Actuarial Science, University of Waterloo , Waterloo , Ontario , Canada
2 Department of Pathology and Laboratory Medicine, University of California, Davis , Sacramento , CA , United States of America
Maguitman Ana
Electronic publication date: 2024 Feb 28
Publication date: 2024
Volume: 10
Electronic Location ID: e1888
Received 2023 Oct 6; Accepted 2024 Jan 29
Copyright: ©2024 Yang et al.
Copyright year: 2024
Copyright holder: Yang et al.
License: This is an open access article distributed under the terms of the Creative Commons Attribution License, which permits unrestricted use, distribution, reproduction and adaptation in any medium and for any purpose provided that it is properly attributed. For attribution, the original author(s), title, publication source (PeerJ Computer Science) and either DOI or URL of the article must be cited.
License URL: https://creativecommons.org/licenses/by/4.0/

Keywords: Natural language processing, NLP, Transformer, BERT, Kidney, Renal, Pathology, Transplant, Language model

Funding: The Canadian Social Sciences and Humanities Research Council (SSHRC) (PI: Schonlau) 435-2021-0287 We received funding from grant 435-2021-0287 from the Canadian Social Sciences and Humanities Research Council (SSHRC) (PI: Schonlau). The funders had no role in study design, data collection and analysis, decision to publish, or preparation of the manuscript.

==============================
Background

Pathology reports contain key information about the patient’s diagnosis as well as important gross and microscopic findings. These information-rich clinical reports offer an invaluable resource for clinical studies, but data extraction and analysis from such unstructured texts is often manual and tedious. While neural information retrieval systems (typically implemented as deep learning methods for natural language processing) are automatic and flexible, they typically require a large domain-specific text corpus for training, making them infeasible for many medical subdomains. Thus, an automated data extraction method for pathology reports that does not require a large training corpus would be of significant value and utility.

Objective

To develop a language model-based neural information retrieval system that can be trained on small datasets and validate it by training it on renal transplant-pathology reports to extract relevant information for two predefined questions: (1) “What kind of rejection does the patient show?”; (2) “What is the grade of interstitial fibrosis and tubular atrophy (IFTA)?”

Methods

Kidney BERT was developed by pre-training Clinical BERT on 3.4K renal transplant pathology reports and 1.5M words. Then, exKidneyBERT was developed by extending Clinical BERT’s tokenizer with six technical keywords and repeating the pre-training procedure. This extended the model’s vocabulary. All three models were fine-tuned with information retrieval heads.

Results

The model with extended vocabulary, exKidneyBERT, outperformed Clinical BERT and Kidney BERT in both questions. For rejection, exKidneyBERT achieved an 83.3% overlap ratio for antibody-mediated rejection (ABMR) and 79.2% for T-cell mediated rejection (TCMR). For IFTA, exKidneyBERT had a 95.8% exact match rate.

Conclusion

ExKidneyBERT is a high-performing model for extracting information from renal pathology reports. Additional pre-training of BERT language models on specialized small domains does not necessarily improve performance. Extending the BERT tokenizer’s vocabulary library is essential for specialized domains to improve performance, especially when pre-training on small corpora.

Introduction

Pathology reports contain crucial diagnostic information, and so do renal pathology reports. Collaborating with the pathology laboratory at the University of California, Davis, we were interested in the diagnostic information presented in their electronic renal pathology records. However, pathology reports are often in an unstructured text format. It is difficult to directly inquire and retrieve information from such unstructured texts. To address this issue and to process unstructured kidney transplantation data, Zubke, Katzensteiner & Bott (2020) created an integration tool based on Microsoft SSIS to transform the data into a structured format. However, converting unstructured language data into a structured, queryable format is labor-intensive and requires pre-determining what set of features will be queried. In recent years, pre-trained language models such as Bidirectional Encoder Representations from Transformers (BERT) (Devlin et al., 2019) have provided a far more flexible and robust system for searching and querying unstructured language data. Such language models based on transformers (Vaswani et al., 2017) have been successfully applied to numerous subject-matter domains (Ahne et al., 2022; Wu et al., 2021; Binsfeld Gonçalves et al., 2022), but typically require a large amount of domain-specific text data for training. Our goal was to develop a pre-trained language model for automatic extraction of information from clinical reports of kidney transplants. To be more specific, we were most interested in two questions: “What kind of rejection does the patient show?” and “What is the grade of interstitial fibrosis and tubular atrophy (IFTA)?” We tested our model on two tasks, classification and information retrieval (IR), to query the reports provided by the pathology laboratory at the University of California, Davis. The reports contained diagnostic information as well as descriptive information regarding the light, immunofluorescence, and electron microscopy findings. In some cases, a comment section that summarizes and interprets the findings was also present.

The experiments conducted pre-training on the given renal pathology reports. Pre-training involves training a language model on a large amount of text prior to considering the specific application of interest. Pre-trained language models have had tremendous success in recent years. In 2019, an attention-based pretrained NLP model, BERT (Devlin et al., 2019), was released by Google. BERT has achieved state-of-the-art performance on the General Language Understanding Evaluation (GLUE) benchmark (Wang et al., 2018), which includes named entity recognition (NER), question and answering (QA) and sentiment classification tasks. For general-purpose NLP tasks, BERT is considered a leading choice.

BERT has been widely applied in the medical domain. For example, Lu et al. (2021), Li et al. (2019) leveraged BERT on unstructured patient-reported records and electronic health notes; Lee, Kang & Yeo (2021) developed a recommendation chatbot for medical specialties using BERT.

Researchers could also pre-train BERT on a medical domain-specific corpus to improve the predictive performance on the task. BioBERT (Lee et al., 2019) is a language model for biomedical language understanding derived from BERT. It was further pre-trained on the PubMed abstracts with 18 billion words. BioBERT benefits from the pre-training process and beats BERT on multiple bio-NLP tasks such as biomedical NER, biomedical relation extraction (RE), and biomedical QA. Later, Alsentzer et al. (2019) proposed Clinical BERT, which is a language model for electronic medical records (EMR). Clinical BERT builds on BioBERT and is further tuned on the EMR notes of the Medical Information Mart for Intensive Care (MIMIC-III) dataset (Johnson et al., 2016) which contains about 60,000 data points. It has been shown that Clinical BERT achieves better results on biological NLP tasks compared to the so called “vanilla” BERT (Alsentzer et al., 2019).

When data are available for a specific clinical subdomain of interest, we can further pre-train Clinical BERT to adapt to that specific clinical subdomain. For example, caBERTnet (Mitchell et al., 2022) is a question-and-answer (QA) system based on Clinical BERT but further pre-trained on the Moffitt (Mitchell et al., 2022) pathology report dataset, which contains 276K reports with 196M words from Moffitt Cancer Center. The authors demonstrated that caBERTnet is superior to Clinical BERT on cancer pathology reports.

We followed a similar procedure and further pre-trained Clinical BERT but for the subdomain of renal pathology reports though, compared to caBERTnet (880M words), we had a much smaller corpus (1.5M words) available to do so. We call the resulting model ‘Kidney BERT’.

However, BERT uses a vocabulary size of only about 30k words. If documents contain words that are not part of the BERT vocabulary, so-called ‘out-of-bag’ (OOB) words, they are tokenized into sub-words pieces. For example, the word “interstitial” will be parsed into frequent sub-words “inter”, “##st”, “##iti”, and “##al” first, and then tokenized into vectors. Both BioBERT and Clinical BERT use the same tokenizer as BERT, which limits their capability to process the OOB complex medical terms. In addition, we noticed that six of the keywords in our two pre-defined study questions (“interstitial”, “fibrosis”, “tubular”, “atrophy”, “T-cell”, and “antibody”) are not contained within the default tokenizer vocabulary, meaning they would be split into sub-words that are unlikely to capture their meaning without significant re-training. We extended the Clinical BERT tokenizer to include these six additional keywords and pre-trained again on our corpus to obtain a new model we call ‘extended Kidney BERT’ (exKidneyBERT).

Method

In this section, we first introduce our dataset (Dataset). Next, we describe how we developed Kidney BERT by pretraining Clinical BERT (BERT Pre-training and Kidney BERT). Then we explain how we extended the vocabulary used in Clinical BERT by six keywords and pre-trained again to obtain exKidneyBERT (Extending Vocabulary for exKidneyBERT). Finally, we lay out how we fine-tuned our models for information retrieval and classification tasks (Fine-tuning BERT models for IR and Classification). The code for this project was open-sourced and can be accessed through this GitHub repository: https://github.com/TianchengY/exKidneyBERT. Portions of this text were previously published as part of the author’s thesis (Yang, 2022).

Ethical consideration

The study has obtained approval from the ethics board at the University of Waterloo, Canada (#4301). The ethics board is the Canadian equivalent of an IRB. The study data are anonymous and deidentified. The Institutional Review Board at University of California at Davis has waived the need for consent from participants of the study (Approval Number: 940174-1).

Dataset

The renal transplant pathology reports used in this study were obtained from the electronic medical records of the University of California, Davis. The pathology reports for transplant kidney biopsy cases consisted of unstructured text for the diagnosis as well as light, immunofluorescence, and electron microscopy results as described by the pathologist. Each report contains the following sections: Diagnosis, Tissues, Gross Description, and Microscopic Description.

For the classification task, the ground truth came from the diagnosis section, which was provided by the pathologist, but it was masked so that the model would not have access to it. For the information retrieval problem, we went through all reports and recorded the desired answers manually. The training target was to match the extracted answers.

There are 3,428 pathology reports in total. Among all the information in the reports, we were interested in the cases with rejection and the cases with interstitial fibrosis and tubular atrophy (IFTA). There are two major types of rejection for patients after kidney transplant in our corpus: T-cell-mediated rejection (TCMR) with 107 positive samples, and antibody-mediated rejection (ABMR) with 123 positive samples (Loupy et al., 2017). The pathology reports classify IFTA into five classes of severity: severe (112 samples), moderate (361 samples), mild (1,120 samples), minimal (347 samples), and absent/insignificant (1,233 samples). We define a sixth class (256 samples) as “unclassified”, meaning the report contains no corresponding information. We used 20% of the data for evaluation. Our goal is to extract parts of sentences or phrases from the report which best describe the condition of rejection and IFTA.

For the classification task, we focused on the content in the Microscopic Description section since it includes the most detailed descriptions of the biopsy. To increase the difficulty of the classification task, we removed any text explicitly related to the task to avoid showing the correct answer in the input text (e.g., the model would need to infer that severe rejection is present and classify accordingly even without seeing the words ‘severe’ or ‘rejection’ anywhere in the text). An example of the input text for the classification task is shown in Table 1. For the IR task, along with the text in the Microscopic Description section, we also added the section of the report comments as a part of the input text since they contain the description of the rejection cases explicitly, and, in this case, we expect the language model to retrieve the answer from the given text. Table 2 shows an example of the input text for the IR task.

Table 1 An illustrative example of the text used for the classification task.

Italicized text was removed during training.

Microscopic Description: The following findings are based on hematoxylin and eosin (H&E), periodic acid-Schiff (PAS), and Masson trichrome-stained sections. The specimen submitted for light microscopic evaluation consists of cortical tissue with at least 35 glomeruli. No segmentally or globally sclerosed glomeruli are seen. The glomeruli demonstrate focal mild mesangial widening. The glomerular capillary walls are of normal thickness and contours. Patchy moderate inflammation is noted associated with scattered moderate tubulitis. The inflammation consists predominantly of mononuclear leukocytes with coms plasma cells and only rare eosinophils. The arteries and arterioles show focal mild hyalinosis. No endotheliitis or peritubular capillaritis is identified.	

Table 2 An illustrative example of the text used for the IR task.

Bold texts are the expected answers. Italicized text was removed during training.

Comments: The biopsy shows interstitial inflammation (i2) consisting of mostly mononuclear leukocytes. Tubulitis (t2) is readily identified in the areas with infiltrating inflammatory cells. These findings support the diagnosis of acute T-cell mediated rejection (IA).	
Microscopic Description: The following findings are based on hematoxylin and eosin (H&E), periodic acid-Schiff (PAS), and Masson trichrome-stained sections. The specimen submitted for light microscopic evaluation consists of cortical tissue with at least 35 glomeruli. No segmentally or globally sclerosed glomeruli are seen. The glomeruli demonstrate focal mild mesangial widening. The glomerular capillary walls are of normal thickness and contours. Patchy moderate inflammation is noted associated with scattered moderate tubulitis. The inflammation consists predominantly of mononuclear leukocytes with coms plasma cells and only rare eosinophils. Mild interstitial fibrosis and tubular atrophy are present (∼10%). The arteries and arterioles show focal mild hyalinosis. No endotheliitis or peritubular capillaritis is identified.	

BERT pre-training and kidney BERT

BERT is a language model consisting of a stack of 12 (BERT-base model) or 24 (BERT-large model) transformer encoder layers containing bi-directional self-attention heads  (Wu et al., 2016). BERT is pre-trained via two unsupervised tasks, masked language modeling and next sentence prediction, on the BooksCorpus (Zhu et al., 2015) and English Wikipedia data. In the masked language modeling stage, 15% of the words in the text were replaced by a special token “[MASK]” to let the model learn and predict the masked word based on the context. More specifically, among the words selected for masking, only 80% of them were replaced by the special mask token. 10% of them are replaced with a random token and the remaining 10% remain the same. Originally, to let the model learn the relationship of sentences, BERT also leveraged a next sentence prediction task. Two sentences are concatenated together by a special token “[SEP]”. 50% of the time, the second sentence is the actual next sentence, and the rest of the time it is chosen randomly. However, in the latest research (Liu et al., 2019), next sentence prediction was found not to be important for improving BERT’s performance.

The prevailing belief (Bear Don’t Walk IV et al., 2021; Tamborrino et al., 2020) suggests that further pre-training on task-specific domains will help improve BERT’s performance on tasks within those domains, and both BioBERT and Clinical BERT took advantage of this sequential pre-training process (Lee et al., 2019; Alsentzer et al., 2019). CaBERT further pre-trained Clinical BERT with Moffitt pathology reports. They simply masked 15% of the words in the Moffitt dataset to a special token “[MASK]”, and then trained the language model to predict these words (Mitchell et al., 2022). While the performance of pre-trained caBERT on the specific downstream tasks of interest was better than when just fine-tuning Clinical BERT, the performance on other tasks with more general corpora such as SQuAD (Rajpurkar et al., 2016) and BioASQ (Balikas et al., 2015) decreased.

We suspect that there are tradeoffs in the pre-training process that depend on the size of the available dataset and choice of downstream task (Sorscher et al., 2022; Wang, Panda & Wang, 2023; Kaplan et al., 2020). Our dataset is much smaller than that used for caBERT: our data contain 3.4K reports with approximately 1.5M words; caBERT is based on 276K reports with 196M words (Mitchell et al., 2022). As a result, we still use the pre-training process suggested by the caBERT authors on our renal pathology reports but conduct an ablation study to determine whether the additional pre-training step adds value.

Extending vocabulary for exKidneyBERT

Both BioBERT and Clinical BERT use WordPiece tokenization (Wu et al., 2016). In our two pre-defined questions for the IR tasks, we found six keywords that are not contained in the original BERT vocabulary and would therefore be parsed into sub-words: “interstitial”, “fibrosis”, “tubular”, “atrophy”, “T-cell”, and “antibody”. We extended the tokenizer to include these six keywords. Also, we needed to extend the embedding layer’s dimensionality from 28,996 to 29,002 to match the newly added words. We extended the tokenizer with additional words found in the two questions. We did not extend the tokenizer with additional words in the reports because (1) these six words contain the most important information needed for the model to locate the answers; (2) extending a lot of words to the tokenizer may affect the pre-trained representations for the existing vocabulary. Since the model does not have any knowledge of the newly added six words, we again applied the same pre-training procedure as above to Clinical BERT (but now using the extended tokenizer) and obtained a new language model we call ‘extended Kidney BERT’ (exKidneyBERT).

Fine-tuning BERT models for IR and classification

Figure 1 shows the architecture we exploited for information retrieval (IR) by using BERT models. For each input, we concatenated “What kind of rejection does the patient show?” or “What is the grade of interstitial fibrosis and tubular atrophy?” to the microscopic description section of the reports together with the special token “[SEP]”. We also added the special token “[CLS]” to the beginning of the concatenated text to follow the BERT usage convention. On top of each BERT model, we added a linear layer as an IR span classifier on the output embedding of BERT. The linear classifier layer is fine-tuned simultaneously with BERT. During fine-tuning, the model predicts a start vector S and an end vector E. The probabilities of each word to be the start and end of the answer are the outputs of vectors S and E after softmax (Bridle, 1989) is applied using the formulas pSi=eSi∑jeSj and pEi=eEi∑jeEj. Next, we applied cross-entropy loss (Cox, 1958) to calculate the gradients: −∑∀y ˆ1X,y ˆlogPy ˆ|X

where 1X,y ˆ is the binary indicator for whether the predicted label y ˆ matches the ground truth label for input X, and Py ˆ|X are the probabilities of the outputs from the softmax. We then updated the parameters of BERT and the classification layer through backpropagation. The words with the maximum probability are chosen as the start and end of the answer text span. If the position of the end word is before that of the start word, then “no information” will be predicted as the output.

In addition to IR, we also tried to use BERT models on the classification task for questions with multiple categories as expected answers. Figure 2 describes the architecture for these tasks. Similar to the IR model, the classification model also exploits a linear classifier layer on top of the BERT models. However, this time we only use the output embedding corresponding to the special token “[CLS]” as the input of the classifier, and then the outputs of the classifier are converted into probabilities through a softmax. The category with the maximum probability is chosen as the final output. Cross-entropy loss is once again used as the loss function. We used HuggingFace transformers (Wolf et al., 2020) as the BERT framework.

Figure 1 Architecture of kidney BERT for the IR task.

Figure 2 Architecture of kidney BERT for the classification task.

Results

After introducing the metrics used for evaluating model performance (Metrics), we report on three results. First, we trained the models on rejection cases only (Training on a portion of reports—rejection cases). Second, we trained the BERT models on all renal pathology reports for the classification tasks (Training on all reports—Classification). Third, we trained the BERT models on all the reports for the IR tasks (Training on all reports—IR).

Metrics

For the first question, “What kind of rejection does the patient show?”, we labelled the text span manually from the reports. A typical answer to the question is “No evidence of acute antibody-mediated rejection”. Since the answers are quite long, we measured the overlap between the predicted text span and the ground truth answer. We calculated the overlap ratio of how much the two text spans overlap on a character-level and word-level, respectively. Measuring character-level overlap ratio could accommodate minor spelling changes and misspellings. If two strings have high character-level overlap, it indicates a strong similarity, even if there are minor differences in spelling. For the second question, “What is the grade of interstitial fibrosis and tubular atrophy?”, since the answers are one-word or two-word phrases, we only counted the prediction results which exactly matched the ground truth phrases. In this case the F1-score was used as the performance metric.

Training on a portion of reports—rejection cases

At the beginning, we focused only on the 242 reports with the rejection cases. Of these, 87 contain positive examples for TCMR. For simplicity, we converted the IR problem into a binary classification task (rather than predicting a text answer). The task is to predict whether the patient’s biopsy shows TCMR based on the report. For the two baseline models, we froze the parameters of Clinical BERT and used (separately) logistic regression and a linear neural network as a classifier with the embedding of the output sentence of BERT as their input. Finally, we fine-tuned a third version of Clinical BERT, this time with the weights unfrozen and a single-layer dense neural network as the classification head. Table 3 shows the results of the three models. We can see that by fine-tuning the classifier and Clinical BERT together, both overall accuracy and F1-score of the positive samples increased substantially.

Table 3 Classification results for freezing Clinical BERT vs. fine-tuning Clinical BERT on the small TCMR sample.

Acc refers to accuracy. Log.Reg. refers to logistic regression. DNN stands for dense neural network.

Model	Overall
Acc	Positive
precision	Positive
recall	Positive
F1-score	
Frozen clinical BERT+Log.Reg.	0.78	0.75	0.23	0.35	
Frozen clinical BERT+DNN	0.88	0.77	0.77	0.77	
Fine-tuned clinical BERT+DNN	0.92	0.87	0.83	0.85	

Training on all reports–classification

In further experiments, we extended the dataset to all 3.4K reports. Analogous to the transfer learning process used for caBERT (Mitchell et al., 2022), we randomly selected and masked 15% of the words in all the reports and trained the Clinical BERT to predict those replaced words. From this pre-training process, we obtained Kidney BERT, our first, novel language model for renal pathology reports. We then extended the tokenizer of Clinical BERT by the six keywords listed above and redid the same pre-training procedure as for Kidney BERT on the 3.4K reports to obtain our second novel model, exKidneyBERT. We fine-tuned all the BERT models in the pre-training chain (see the X-axis in Fig. 3) including the vanilla-cased base BERT, BioBERT, Clinical BERT, Kidney BERT, and exKidneyBERT on the rejection classification tasks. We also added IFTA grade classification as a second task to further compare the performance of these models. The results are shown in Table 4.

Figure 3 BERT models’ results of the IR task for ABMR.

Table 4 Classification results for fine-tuning BERT models on the full dataset.

CLS means classification task.

Model	Task	Overall
Acc.	Positive
precision	Positive
recall	Positive
F1-score	
BERT	Rej. CLS	0.945	0.000	0.000	0.000	
BioBERT	Rej. CLS	0.953	0.697	0.447	0.515	
Clinical BERT	Rej. CLS	0.977	0.923	0.632	0.750	
Kidney BERT	Rej. CLS	0.977	0.867	0.684	0.765	
exKidneyBERT	Rej. CLS	0.978	0.811	0.789	0.800	
Model	Task	Overall
Acc.	Weighted
precision	Weighted
recall	Weighted
F1-score	
BERT	IFTA CLS	0.702	0.690	0.697	0.679	
BioBERT	IFTA CLS	0.731	0.729	0.731	0.726	
Clinical BERT	IFTA CLS	0.733	0.734	0.732	0.729	
Kidney BERT	IFTA CLS	0.730	0.731	0.722	0.724	
exKidneyBERT	IFTA CLS	0.714	0.710	0.712	0.708	

Training on all reports–IR

In addition to classification tasks, we also considered information retrieval (IR) tasks. We manually tagged the desired answer phrases corresponding to ABMR and TCMR rejection in each report for the question “What kind of rejection does the patient show?”. For the question related to IFTA, “What is the grade of interstitial fibrosis and tubular atrophy?”, we tagged any mention of the six outcome classes as expected answers. We fine-tuned all the BERT models again, each with an IR head attached. For the IR tasks, the question and text are concatenated as the model input. Table 5 and Figs. 3–5 show the results.

Table 5 IR results for fine-tuning BERT models on the full dataset.

Overlap Ratio char and Overlap Ratio word refer to the average overlap length between the predicted answers and the expected answers divided by the length of the expected answers, at character-level and word-level, respectively. Exact match rate refers to the proportion of perfect matches between the predicted answers and the expected answers.

Model	Task	Overlap	Overlap	
		Ratio char	Ratio word	
BERT	ABMR IR	0.442	0.616	
BioBERT	ABMR IR	0.519	0.667	
Clinical BERT	ABMR IR	0.363	0.461	
Kidney BERT	ABMR IR	0.363	0.461	
exKidneyBERT	ABMR IR	0.604	0.833	
BERT	TCMR IR	0.494	0.653	
BioBERT	TCMR IR	0.494	0.653	
Clinical BERT	TCMR IR	0.494	0.653	
Kidney BERT	TCMR IR	0.494	0.653	
exKidneyBERT	TCMR IR	0.664	0.792	
Model	Task	Exact match rate		
BERT	IFTA IR	0.942		
BioBERT	IFTA IR	0.956		
Clinical BERT	IFTA IR	0.947		
Kidney BERT	IFTA IR	0.950		
exKidneyBERT	IFTA IR	0.958		

Figure 4 BERT models’ results of the IR task for TCMR.

Figure 5 BERT models’ results of the IR task for IFTA.

We also found that exKidneyBERT overcomes some of the common failure modes other models have. For example, in the IR task of ABMR, exKidneyBERT performed better on retrieving the information containing the term “antibody-mediated” correctly while other models could not. A possible reason is that the tokenizer of exKidneyBERT will parse the term “antibody-mediated” into “antibody”, “-”, and “mediated”, while others will parse it into “anti”, “body”, “-”, “mediated”. A similar case happened in the IR tasks of TCMR where exKidneyBERT performed better on the term “T-cell-mediated” than other models.

Discussion

Principal results

Our study shows the importance of extending the vocabulary of language models to include keywords found in the queries used for IR tasks. In addition, we compared the performance differences of various BERT models on our kidney transplant pathology report dataset.

First, we found that by extending the tokenizer with the six keywords from the questions of the IR tasks, exKidneyBERT outperforms the other BERT models. We compared five BERT models in total. BioBERT was pre-trained with the PubMed corpus on the cased base BERT model. Clinical BERT was pre-trained with the MIMIC-III dataset on BioBERT. We created Kidney BERT by pre-training with our renal pathology data on Clinical BERT and we developed exKidneyBERT by extending the tokenizer of Clinical BERT with six keywords in the two questions in our IR tasks and pre-training with our data on Clinical BERT.

In the classification task of rejection case, exKidneyBERT performed best on both overall accuracy and F1-score of positive samples. But for the classification task of IFTA, exKidneyBERT performs the second worst and the result of BioBERT beats others. This is not surprising since, as specified in the task design, we removed the sentences that contained the six keywords from the reports to make the task more difficult. This result confirms that exKidneyBERT’s improved performance was specifically due to it making use of its extended vocabulary.

For the IR tasks of ABMR and TCMR, exKidneyBERT outperforms the other four BERT models on both overlap ratio at character level and word level. Notice that in the TCMR case, exKidneyBERT achieved substantially better results (0.664 and 0.792 on overlap ratio at character level and word level, respectively) than the other four BERT models (all had 0.494 on characters’ overlap ratio and 0.653 on words’ overlap ratio). For the IR tasks with IFTA, exKidneyBERT achieved the best result among all five BERT models. Unlike the classification tasks, the input text in the IR tasks contains the sentences include the six keywords, which allows exKidneyBERT to benefit from the extended six keywords.

Second, we performed an ablation study to determine which modeling components contributed to the performance increase. We found that performing masked language model pre-training on increasingly small domain-specific text corpuses without extending the vocabulary did not improve the performance in our domain. The success of previous language models like Clinical BERT and CancerBERT appeared to suggest that when adapting BERT to a particular domain, the BERT model will benefit from pre-training with the domain-specific corpus. We found that pre-training on a small domain-specific corpus for renal pathology reports is ineffective on its own. On the classification task, the results for Kidney BERT are the same as those for Clinical BERT in overall accuracy, and only 0.015 higher in F1-score of positive samples (see Tables 4 and 5). On the IR tasks, the results for Clinical BERT and Kidney BERT are the same on the rejection tasks, and the exact match rate of IFTA task with Kidney BERT is only 0.003 higher than Clinical BERT. Fine-tuning the entire model (rather than using frozen parameters with a fine-tuned task-specific head as is often done in pre-training setups) was beneficial based on the results shown in Table 3.

Third, we found that for the specific subdomain of our renal pathology reports, BioBERT performed better than Clinical BERT. We fine-tuned the BERT models on five tasks in total and, in all tasks except for the rejection case of the classification task, BioBERT outperformed Clinical BERT. This may be because Clinical BERT was pre-trained on a different subdomain than our dataset while BioBERT was pre-trained on a more general domain that may have provided better coverage of our dataset.

Limitations

First, exKidneyBERT was only designed to answer the two pre-defined questions. For exKidneyBERT, we extended the tokenizer with the six keywords from the two questions of the IR tasks we wanted to resolve. As always, if we desire to solve other IR tasks, we need to train new models (with potentially more keywords) for them.

Second, the dataset we used is small compared to the other BERT models. Google BERT was pre-trained on 3.3 billion words, BioBERT was pre-trained on 18 billion words, Clinical BERT was pre-trained on 880 million words, and the dataset we used for pre-training only contains 1.5 million words. Pre-training on a larger dataset may further increase the performance of Kidney BERT and exKidneyBERT; however, data availability is limited in many domains including ours, and in order to investigate pre-training when working with scarce data, we carry out our evaluations on a small dataset.

Comparison with prior work

We followed a similar unsupervised pre-training procedure as CancerBERT (Mitchell et al., 2022) when developing Kidney BERT. Following this procedure, we initialized the model parameters from Clinical BERT and randomly selected 15% of the words and replaced them with a special token “[MASK]” and then trained the model to predict the masked tokens. In addition, we added the six keywords from the questions of the IR tasks to the BERT tokenizer and repeated the same pre-training procedure as CancerBERT on our dataset to create exKidneyBERT. We found that on our kidney transplant pathology reports, exKidneyBERT performs better than Kidney BERT, and is therefore an improvement compared to the procedure of CancerBERT. We also benchmarked our proposed methods against several other BERT variants as laid out in the results.

Conclusions

We have made two primary contributions. First, we developed exKidneyBERT, a language model with an extended vocabulary of six keywords specific to renal pathology reports and showed that exKidneyBERT outperformed existing models in the IR tasks. Second, while designing the model, we found that contrary to conventional wisdom (Bear Don’t Walk IV et al., 2021; Tamborrino et al., 2020), pre-training is not “all you need”. In particular, we found that in our renal pathology dataset, BioBERT performed better than Clinical BERT on some of the tasks, even though Clinical BERT has additional pre-training over BioBERT. In addition, we conducted an ablation study and found that Kidney BERT (without extended vocabulary) did not outperform other models even though it was further pre-trained on our renal pathology dataset. Our insights suggest that two factors must be met for pre-training in similar challenges (e.g., other medical fields) to be successful: (1) the training corpus needs to be well-aligned with the subdomain the model will be used on, and (2) especially when pre-training on narrow subdomains with limited data, the model vocabulary needs to be extended to explicitly include technical terms relevant to that subdomain. These two findings would be helpful for people applying BERT-based language models, e.g., BioBERT, Clinical BERT, on narrow medical subdomains with OOB technical terms. Future research should confirm the importance of out-of-bag words in highly specialized domains.

Supplemental Information

Supplemental Information 1 Raw reports, processed texts, and labels

Supplemental Information 2 Code in Python

This article is based on the first author’s thesis (Yang, 2022).

Additional Information and Declarations

Competing Interests

Author Contributions

Ethics

Data Availability

The authors declare there are no competing interests.

Tiancheng Yang conceived and designed the experiments, performed the experiments, analyzed the data, performed the computation work, prepared figures and/or tables, authored or reviewed drafts of the article, and approved the final draft.

Ilia Sucholutsky conceived and designed the experiments, analyzed the data, authored or reviewed drafts of the article, and approved the final draft.

Kuang-Yu Jen performed the experiments, analyzed the data, authored or reviewed drafts of the article, and approved the final draft.

Matthias Schonlau conceived and designed the experiments, analyzed the data, authored or reviewed drafts of the article, and approved the final draft.

The following information was supplied relating to ethical approvals (i.e., approving body and any reference numbers):

The Institutional Review Board at the University of California, Davis, approved this research and waived the need for informed consent.

The following information was supplied regarding data availability:

The data and codes are available in the Supplemental Files.

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
