# Peer review of "exKidneyBERT: a language model for kidney transplant pathology reports and the crucial role of extended vocabularies"

_PeerJ Computer Science, doi:10.7717/peerj-cs.1888_

## Round 0.1 · original submission · Major Revisions

The reviewers recommend reconsideration of your manuscript following major revision. I invite you to resubmit your manuscript after addressing the comments made by the reviewers. Make sure that the revised manuscript includes the additional results requested by the reviewers. In particular, include an analysis of the underlying reasons behind the choices made, perform a comparison analysis against existing approaches, discuss the limitations of the proposed approach, and add suggestions for future work.

I hope you can complete the recommendation changes in a revision of your article.

**Language Note:** PeerJ staff have identified that the English language needs to be improved. When you prepare your next revision, please either (i) have a colleague who is proficient in English and familiar with the subject matter review your manuscript, or (ii) contact a professional editing service to review your manuscript. PeerJ can provide language editing services - you can contact us at copyediting@peerj.com for pricing (be sure to provide your manuscript number and title). – PeerJ Staff

Reviewer 1 ·

Basic reporting

The authors present a re-training and modification to the Clinical BERT architecture to improve the performance of the BERT model on information retrieval and classification tasks performed in electronic medical records of renal pathology reports.

What seems interesting about this work is the explicit inclusion of keywords to the language model and the effect on the IR and classification performance. However, there are some aspects that the authors should consider:

I noticed that the manuscript is strongly based on a previous master's degree thesis; I suggest summarizing and paraphrasing the thesis to be punctual with the hypothesis question that you want to answer.

The introduction section seems a little messy. For example, in lines 59-61, the first sentence tries to explain some background to the problem, and the next sentence presents some state-of-the-art solutions. I suggest the authors re-organize the text in some order, for example, introduce the topic, describe some background, state the problem of interest, a little discussion of the state-of-the-art, and state the objectives.

There are some sentences in the “Introduction” section that would be better presented in other sections, for example:

In the “Method” section, the first sentence (lines 40-43) seems more like an argument for the “Objective” section. In the “Methods” section just present your computational proposal.

Lines 105-108: This sentence should be presented in the conclusion rather than the introduction.

Line 37: The sentence should be started with a capital letter.

Lines 143-144: Could be appended to the previous paragraph.

Lines 109-112: This sentence contains the same information as in line 79. If this sentence does not offer more information, please remove it. Also, line 79 presents inconsistencies regarding the year of the BERT presentation.

Line 173 and 177: Table order and citing should be in numerical ascending order. Please amend.
Lines 165-166: Do not start a sentence with a number.

Fig 3-5 should indicate the metric used as an axis label. Also, according to the authors' guidelines, they must contain a Title.

Experimental design

Line 156-157: The author makes use of a set of rule-based heuristics to extract relevant sentences from the EMR. To have reproducibility, it is important to indicate every step of the methodology, including the set of rule-based heuristics and how they were obtained. Additionally, considering that an important part of the contribution of this work is the information retrieval model. Could the authors explain whether this heuristic affects the performance of the IR proposal? I wonder this because the idea of IR systems is to work on raw data.

Lines 165-166: I understand that the authors split the dataset into 80%-20% for training and testing. However, has the training-validation process been executed several times? considering that the dataset was small, wouldn't it be a good idea to with the remaining 80% using a k-fold cross-validation strategy?

Lines 202-203: The authors suspect that “there are tradeoffs in the pre-training process that depend on the size of the available dataset”. I suggest that you look for evidence on the state-of-the-art that supports these claims.

Lines 268-269: ¿How character-level overlap ratio could be a good metric for the English language? please explain.

About the proposal code, with the code shared as a notebook and with no explicit access point, it is difficult to follow it and identify the modifications made by the authors. I suggest rewriting the code in a way that could be used by any person interested in it.

Validity of the findings

The most important conclusion of the authors claims that the explicit inclusion of keywords in the model language vocabulary plays an important role in the performance of the models. Even if this is evident in the experiments with this dataset it would be interesting to be proven with other datasets in order to generalize this behavior.

Regarding training with domain-specific corpus, and the effect of the corpus size on classification task; it would be interesting to contrast your findings with other works in the literature since this conclusion has been found in other works. Please, perform a deeper search in this regard. However, what is interesting and would be an important contribution is the possibility of leveraging the language model performance by explicitly adding important words, but as I stated before, deeper experimentation must be done.

Additional comments

no comment

Reviewer 2 ·

Basic reporting

No comment

Experimental design

No comment

Validity of the findings

No comment

Additional comments

Abstract
• The abstract neatly outlines the study's aims and achievements. Adding a brief note on the methodological innovations, like enhancing the BERT model's vocabulary, would make it even more informative.
Introduction
• Clarifying the existing research gap, especially regarding how current models fall short in processing specialized vocabularies in renal pathology reports, would make the introduction stronger. How does this new model stand out from the existing ones in this aspect?
• More emphasis on the challenges of handling complex medical terms and jargon in kidney pathology reports by existing models would be insightful.
• It would be great if the introduction could delve deeper into the impact of enriching the tokenizer with specific medical terms before pre-training. Highlighting how this strategy empowers the model to accurately interpret and process unique medical terminologies in kidney pathology reports would be very useful.
Method
• The method section is well-organized and thoroughly covers the development of Kidney BERT and exKidneyBERT, from dataset preparation to the intricate stages of model training.
• A more in-depth discussion on why certain keywords were chosen for vocabulary extension and their significance in renal pathology would enhance this section.
Results
• Discussing how these findings could be applied in real clinical settings would greatly add to the paper's practical value.
Discussion
• The discussion does a good job of underscoring the importance of expanded vocabularies in specialized language models and compares various BERT models effectively.
• It would be beneficial to explore how these insights could apply to other medical fields or similar challenges in text analysis.
Conclusions
• Adding suggestions for future research, such as adapting the model for other medical areas or improving it with more extensive datasets, would be a valuable addition to the conclusions.

---

## Round 0.2 · accepted · Accept

Thank you for your contribution to PeerJ Computer Science and for addressing all the reviewers' suggestions. The reviewers are satisfied with the revised version of your manuscript and it is now ready to be accepted. Congratulations!

Reviewer 1 ·

Basic reporting

No comment

Experimental design

No comment

Validity of the findings

No comments

Additional comments

No comments

Reviewer 2 ·

Basic reporting

The authors have addressed all comments and suggestions.

Experimental design

No comment

Validity of the findings

No comment

Additional comments

No comment